# Negative Effect of Reduced NME1 Expression on Recurrence-Free Survival in Early Stage Non-Small Cell Lung Cancer

**DOI:** 10.3390/jcm9103067

**Published:** 2020-09-23

**Authors:** Dohun Kim, Yujin Kim, Bo Bin Lee, Dongho Kim, Ok-Jun Lee, Pildu Jeong, Wun-Jae Kim, Eun Yoon Cho, Joungho Han, Young Mog Shim, Duk-Hwan Kim

**Affiliations:** 1Department of Thoracic and Cardiovascular Surgery, Chungbuk National University Hospital, College of Medicine, Chungbuk National University, Cheongju 28644, Korea; mwille@naver.com; 2Department of Molecular Cell Biology, Sungkyunkwan University School of Medicine, Suwon 16419, Korea; yujin0328@hanmail.net (Y.K.); whitebini@hanmail.net (B.B.L.); jindonghao2001@hotmail.com (D.K.); 3Department of Pathology, College of Medicine, Chungbuk National University, Cheongju 28644, Korea; ojlee@chungbuk.ac.kr; 4Department of Urology, College of Medicine, Chungbuk National University, Cheongju 28644, Korea; leo24fly@chungbuk.ac.kr (P.J.); wjkim@chungbuk.ac.kr (W.-J.K.); 5Department of Pathology, Samsung Medical Center, Sungkyunkwan University School of Medicine, Seoul 06351, Korea; eunyoon.cho@samsung.com (E.Y.C.); joungho.han@samsung.com (J.H.); 6Department of Thoracic and Cardiovascular Surgery, Samsung Medical Center, Sungkyunkwan University, School of Medicine, Seoul 06351, Korea; youngmog.shim@samsung.com

**Keywords:** adjuvant chemotherapy, β-catenin, lung neoplasms, nucleotide-diphosphate kinase, recurrence

## Abstract

This study aimed to understand whether the effect of non-metastatic cells 1 (NME1) on recurrence-free survival (RFS) in early stage non-small cell lung cancer (NSCLC) can be modified by β-catenin overexpression and cisplatin-based adjuvant chemotherapy. Expression levels of NME1 and β-catenin were analyzed using immunohistochemistry in formalin-fixed paraffin-embedded tissues from 425 early stage NSCLC patients. Reduced NME1 expression was found in 39% of samples. The median duration of follow-up was 56 months, and recurrence was found in 186 (44%) of 425 patients. The negative effect of reduced NME1 expression on RFS was worsened by cisplatin-based adjuvant chemotherapy (adjusted hazard ratio = 3.26, 95% CI = 1.16–9.17, *p* = 0.03). β-catenin overexpression exacerbated the effect of reduced NME1 expression on RFS and the negative effect was greater when receiving cisplatin-based adjuvant chemotherapy: among patients treated with cisplatin-based adjuvant chemotherapy, hazard ratios of patients with reduced NME1 expression increased from 5.59 (95% confidence interval (CI) = 0.62–50.91, *p* = 0.13) to 15.52 (95% CI = 2.94–82.38, *p* = 0.001) by β-catenin overexpression, after adjusting for confounding factors. In conclusion, the present study suggests that cisplatin-based adjuvant chemotherapy needs to be carefully applied to early stage NSCLC patients with overexpressed β-catenin in combination with reduced NME1 expression.

## 1. Introduction

Lung cancer is one of the most common causes of cancer-related deaths in the world. Despite recent advances in the early detection and treatment of lung cancer, the prognosis is very poor, partly because of a high rate of recurrence even after curative resection. Approximately half of the patients diagnosed with non-small cell lung cancer (NSCLC) develop recurrence and die of the disease even after curative resection. Adjuvant chemotherapy plays an important role in preventing recurrence following curative resection of lung cancer. A survival benefit of platinum-based adjuvant chemotherapy in NSCLC was confirmed by phase III trials and the Lung Adjuvant Cisplatin Evaluation (LACE) meta-analysis [1,2]. However, some NSCLC patients receiving such adjuvant chemotherapy show no progress in survival. Accordingly, it is critically important to identify biomarkers that can select patients who will not respond well to adjuvant therapy so that an appropriate treatment plan can be provided to patients. Given that occult micro-metastatic cancer cells might be present systemically at the time of surgery, altered expression of metastasis-related genes might be useful as molecular biomarkers to distinguish patients at high risk of recurrence after surgery.

Non-metastatic cells 1 (NME1), also known as NM23-H1, was the first metastasis suppressor discovered by its reduced mRNA transcript levels in a murine melanoma cell line exhibiting high metastatic activity [3]. In addition to its known function as a nucleotide-diphosphate kinase that converts nucleoside diphosphates to nucleoside triphosphates at the expense of adenosine triphosphate (ATP), NME1 is involved in several pathological processes such as motility and metastasis of tumor cells [4]. An inverse relationship between metastatic potential and NME1 expression has been reported in several types of cancers, including non-small cell lung cancer [5,6], melanoma [7], breast cancer [8], hepatocellular carcinoma [9], gastric cancer [10], and colorectal cancer [11]. Transfection of the *NME1* gene into different types of cancer cells has resulted in the inhibition of metastatic properties, including migration, invasion, and colonization [12,13,14,15,16]. *NME1* silencing is known to upregulate β-catenin-dependent TCF/LEF-1 (T-cell factor/lymphoid enhancer-binding factor) transactivation through glycogen synthase kinase (GSK)-3β-independent mechanisms by promoting nuclear translocation of β-catenin [17].

Activation of the canonical Wnt signaling pathway inhibits axin-mediated β-catenin phosphorylation and degradation and allows β-catenin to accumulate in the cytoplasm and then translocate into the nucleus. Nuclear β-catenin forms a stable complex with members of the TCF/LEF transcription factor family and induces the expression of target genes such as *c-MYC* and *CCND1*, and influences the metastatic cascade by regulating the expression of genes such as *AXIN2*, *SNAIL*, *ZEB1*, *COX2*, and *S100A4* [18]. It has been reported that the Wnt/β-catenin signaling pathway is involved in the invasion and metastasis of tumor cells in patients with NSCLC [19,20,21,22]. In addition to the nuclear translocation of β-catenin by *NME1* silencing, Wnt/β-catenin-mediated resistance to cisplatin has been demonstrated in human cancers [23,24]. Based on these reports, we hypothesized that NME1 and the Wnt signal may cooperatively affect patient prognosis and cisplatin treatment.

In this study, we analyzed whether the effect of NME1 on recurrence-free survival (RFS) can be modified by cisplatin-based adjuvant chemotherapy and β-catenin overexpression in early stage NSCLC.

## 2. Materials and Method

### 2.1. Study Population

This was a retrospective study. Formalin-fixed paraffin-embedded (FFPE) tissue specimens stored at room temperature were obtained from 425 patients with pathologic stage I–IIIA NSCLC who had undergone anatomical lung resection with mediastinal lymph node dissection between November 1994 and April 2004 at Samsung Medical Center in Seoul, Korea. Patients with incomplete resection of lung tissue (e.g., positive malignant cell in resection margin) or history of neoadjuvant therapy were excluded from this study. Postoperative follow-up was performed according to a previously described protocol [25]. Information including recurrence, death, and platinum-based adjuvant chemotherapy was obtained from our hospital’s electronic medical records (EMRs) and outside medical records as of 31 July 2018. Thirty-two (7.5%) patients received postoperative adjuvant chemotherapy comprising cisplatin combined with vinorelbine, vinblastine, etoposide, fluorouracil, gemcitabine, pemetrexed, or docetaxel. The chemotherapy regimens were selected by medical oncologists responsible for treatment decisions. This study was approved by the Institutional Review Board of the Samsung Medical Center (2018-04-153), and pre-operative informed consent for the use of samples was obtained from all patients. Pathologic stage was determined according to the guideline of the 7th edition of the tumor-node-metastasis (TNM) staging system maintained by the American Joint Committee on Cancer [26]. Supporting data for this study are available from the corresponding author upon request.

### 2.2. Immunohistochemistry

Tissue microarrays (TMAs) were constructed from paraffin blocks prepared from 425 NSCLC samples. Expression levels of β-catenin and NME1 proteins were analyzed using immunohistochemistry. In brief, serial sections of 4 μM in thickness were cut from TMA blocks, deparaffinized in xylene, and rehydrated through a series of decreasing concentrations of alcohols. Antigens were recovered by heating these sections in 10 mM (pH 6) citrate buffer for 10 min using a pressure cooker. These sections were then incubated with primary antibody β-catenin clone 17C2 (Leica Biosystems, Buffalo Grove, IL, USA) or NME1 clone 4B2 (GeneTex, Irvine, CA, USA) at 4 °C overnight. Immunoreactivity of each primary antibody was detected with Envision™ + peroxidase (Dako, Carpinteria, CA, USA). Antibody-bound peroxidase activity was visualized after incubating with chromogen 3,3′-diaminobenzidine (DAB) at room temperature for 1–5 min. Normal bronchial epithelial cells were used for positive control of staining, and primary antibody was replaced by immunoglobin for negative control. All sections were counterstained with Mayer’s hematoxylin.

### 2.3. Interpretation of Immunohistochemical Staining

Immunohistochemical staining was interpreted by consensus between two authors (O.-J.L. and D.-H.K.) in a double-blinded fashion to minimize inter-rater variability. Samples with a Cohen’s kappa coefficient of less than 0.20 were removed from further analysis. Although immunoreactivity for β-catenin was found in the membrane, cytoplasm, and nucleus, only cytoplasmic staining was assessed for scoring. The expression of NME1 protein in tumor cells was evaluated based on cytoplasmic staining. Cytoplasmic staining of both proteins was semi-quantitatively evaluated using a score calculated by multiplying the intensity score (0, none; 1, weak; 2, moderate; 3, strong) with the proportion score of positive cells (0, absent; 1, 0–10%; 2, 10–50%; 3, 50–80%; 4, >80%). For NME1, its expression was defined as reduced if a composite score was less than two in a tumor. β-catenin expression was considered to be overexpressed in a tumor with a composite score greater than or equal to two. Staining was performed in triplicate and average values of scores were used to determine the expression levels. Cutoff values for the abnormal expression of NME1 and β-catenin were determined considering an internal control consisting of 23 normal lung cores. Representative positive stainings for β-catenin and NME1 expression are shown in adenocarcinoma and squamous cell carcinoma (Figure 1A). Details of the immunohistochemical staining procedure and interpretation for Ki-67 (MKI67) proteins were reported previously [27].

### 2.4. Study Design

Patients were randomly selected without stratification or matching by age. The median duration of follow-up was 56 months. The clinical endpoint of the study was recurrence-free survival (RFS), which was defined as the time from the date of the diagnosis to the first recurrence. Variables such as age, sex, histology, pathologic stage, NME1 expression, and adjuvant chemotherapy were initially considered for the analysis of RFS. FFPE tissue samples were obtained from 425 patients because at least 365 patients were needed for analysis of the effect of NME1 expression on RFS under 2-sided α = 0.05 and β = 0.1 (i.e., 90% power).

### 2.5. Statistical Analyses

To find factors associated with NME1 reduction in NSCLC, chi-square test (or Fisher’s exact test) and Student’s *t*-test (or one-way ANOVA) were used for univariate analyses of continuous and categorical variables, respectively. A linear relationship between two continuous variables was analyzed using Pearson’s correlation coefficient. The prognostic significance of NME1 on RFS was evaluated by Kaplan–Meier survival curves. The difference between two survival curves was assessed using the log-rank test. Variables with *P* ≤ 0.25 in the univariate analysis were included in the multivariate model. Hazard ratios of predictor variables for survival were estimated using the Cox proportional hazards model after controlling for potential confounding factors. The effect of β-catenin expression and adjuvant chemotherapy on NME1 function was analyzed using a stratified Cox proportional hazards model. No replacement was made for missing values. All statistical analyses were two-sided with a type I error of 5%.

## 3. Results

### 3.1. Clinicopathological Characteristics

A total of 425 patients with early stage NSCLC were included in the data analysis without dropout. The mean age of the patients at diagnosis was 61 years (range, 37–82 years), and men accounted for 74% of the cases. Adenocarcinoma and squamous cell carcinoma comprised 46% and 47% of the cases, respectively. Patients at I, II, and IIIA stages accounted for 56%, 43%, and 5%, respectively. The relationship between NME1 expression and clinicopathological characteristics is summarized in Appendix A. NME1 expression was found to be reduced in 165 (39%) of 425 patients. Reduced NME1 expression was not associated with patient’s age, sex, tumor size, or exposure to tobacco smoke. However, reduced NME1 expression was found to have a significantly higher prevalence in squamous cell carcinoma (46%) than in adenocarcinoma (34%), and the difference was statistically significant (*p* = 0.01; Figure 1B).

Postoperative recurrence occurred in 186 (44%) of 425 patients. Patients with reduced NME1 expression had a higher recurrence rate than those without (62% vs. 32%, *p* < 0.0001). β-catenin was overexpressed in 55% of patients, with a higher prevalence in squamous cell carcinoma than that in adenocarcinoma (*p* = 0.005; Figure 1B). Recurrence was found at a high prevalence in patients with reduced NME1 expression but not β-catenin overexpression irrespective of histologic subtypes (Figure 1C).

### 3.2. Reduced NME1 Expression Is Significantly Associated with Poor RFS Irrespective of Histology or Pathologic Stage

Univariate analysis was performed to discover prognostic factors that affect RFS in early stage NSCLC. RFS was negatively associated with reduced NME1 expression but not with β-catenin overexpression and cisplatin-based adjuvant chemotherapy (Appendix A). Patients were stratified according to histology and pathologic stage to analyze whether the relationship between RFS and NME1 expression was modified by histology or pathologic stage. RFS was compared between patients with and without reduced NME1 expression in histologic subtypes. Reduced NME1 expression was significantly associated with RFS (*p* < 0.0001; Appendix A): Five-year RFS rate after surgery was 38% for those with reduced NME1 expression and 68% for those without reduced NME1 expression. Reduced NME1 expression had a negative effect on RFS in adenocarcinoma (*p* < 0.0001; Appendix A) and in squamous cell carcinoma (*p* < 0.0001; Appendix A).

The effect of reduced NME1 expression on RFS was further analyzed based on pathologic stage (Figure 2). The number of patients with stage IIIA NSCLC was five, which was too small to analyze RFS. Therefore, patients with stage IIIA NSCLC were combined with those who had stage IIB NSCLC to analyze RFS. Reduced NME1 expression was significantly associated with poor RFS in stage IA (*p* = 0.0005; Figure 2A), stage IB (*p* = 0.001; Figure 2B), and stage IIA (*p* = 0.01; Figure 2C). It was marginally associated with poor RFS in stage IIB–IIIA (*p* = 0.08; Figure 2D). The relationship between β-catenin overexpression and RFS was also analyzed based on pathologic stage and histology. However, no association was found between the.

### 3.3. Negative Effect of Reduced NME1 Expression on RFS in Patients with Cisplatin-Based Adjuvant Chemotherapy and β-Catenin Overexpression

Cisplatin-based adjuvant chemotherapy was not associated with RFS irrespective of histologic subtypes in univariate analysis. The effect of NME1 or β-catenin on RFS was further analyzed considering cisplatin-based adjuvant chemotherapy. The negative effect of reduced NME1 expression on RFS was worse in patients treated with (Figure 3A) cisplatin-based adjuvant chemotherapy than in those treated without (Figure 3B). Reduced NME1 expression was not associated with β-catenin overexpression in this study. However, it is known that there is a complex interplay between NME1 and β-catenin in a variety of cancers. Therefore, data were further stratified according to β-catenin overexpression. The negative effect of reduced NME1 expression on RFS was much greater in patients with overexpression of β-catenin (Figure 3C) than in those without (Figure 3D).

### 3.4. Multivariate Cox Proportional Hazards Analysis

Multivariate Cox proportional hazards analysis was conducted to measure the effect of reduced NME1 expression on RFS in early stage NSCLC, after adjusting for potential confounding effects of variables. Considering the pathological stage, hazard ratios for RFS ranged from 1.64 to 3.93, after adjusting for patient age, sex, β-catenin expression, adjuvant chemotherapy, and histology (Appendix A). The hazard ratio for RFS in a total of 425 patients was 2.27 (95% CI = 1.70–3.03, *p* < 0.0001) times worse in patients with reduced NME1 expression than in those without (Appendix A). However, the hazard ratio was not associated with age (HR = 1.01, 95% CI = 0.99–1.03, *p* = 0.18), sex (HR = 1.02, 95% CI = 0.71–1.47, *p* = 0.93), adjuvant chemotherapy (HR = 1.03, 95% CI = 0.86–2.23, *p* = 0.89), and β-catenin expression (HR = 0.99, 95% CI = 0.75–1.34, *p* = 0.99).

To test our hypothesis that the effect of NME1 expression on RFS may be affected by Wnt signal and cisplatin treatment, we stratified patients according to adjuvant chemotherapy and β-catenin expression. The negative effect of reduced NME1 expression on RFS was exacerbated by cisplatin-based adjuvant chemotherapy (adjusted hazard ratio = 3.26, 95% CI = 1.16–9.17, *p* = 0.03; Appendix A). For patients who did not receive cisplatin-based adjuvant chemotherapy, the hazard ratio of reduced NME1 expression was increased from 1.89 (95% confidence interval (CI) = 1.21–2.95, *p* = 0.005) to 2.54 (95% CI = 1.66–3.89, *p* < 0.0001; Table 1) by β-catenin overexpression after adjusting for patient age, sex, histology, and pathologic stage. Among patients with β-catenin overexpression who received cisplatin-based adjuvant chemotherapy, patients with reduced NME1 expression were determined to have 15.52 (95% CI = 2.94–82.38, *p* = 0.001; Table 1) times poorer RFS than those without. These results suggest that the negative effect of reduced NME1 expression on RFS may be worsen by cisplatin-based adjuvant chemotherapy and β-catenin overexpression in early stage NSCLC.

### 3.5. Relationship between Ki-67 Labeling Index and Expression of NME1 and β-Catenin

Ki-67 proliferation index was analyzed to investigate whether the effect of NME1 and β-catenin on cisplatin-resistance might be confounded by different cell proliferation activity. Ki-67 proliferation was not significantly different according to abnormal expression of NME1 or β-catenin irrespective of histology (Figure 4A) and cisplatin-based adjuvant chemotherapy (Figure 4B). These observations suggest that NME1 and β-catenin may affect cisplatin-based adjuvant chemotherapy through other mechanisms rather than through any changes in cell proliferation.

### 3.6. The Relationship between NME1 and Nuclear β-Catenin Expression

*NME1* silencing is known to induce a redistribution of β-catenin from the cell surface into the cytoplasm and nucleus by activating the Wnt pathway [17]. To understand whether NME1 expression affects the nuclear translocation of β-catenin, we analyzed the levels of nuclear β-catenin expression (Figure 5A) according to NME1 expression. The levels of nuclear β-catenin expression showed a linear relationship with those of cytoplasmic β-catenin expression (Pearson’s correlation coefficient γ = 0.74, *p* = 0.002; Figure 5B) but not to those of NME1 expression (γ = −0.03, *p* = 0.80; Figure 5C).

## 4. Discussion

The epithelial–mesenchymal transition (EMT) plays a crucial role in promoting metastasis of carcinoma derived from epithelial cells. Tumor cells lose their epithelial characteristics such as cell polarity and gain mesenchymal features such as increased migratory and invasive potentials during EMT. Our data did not show an association of NME1 expression with tumor growth (Figure 4), consistent with a previous study showing that *NME1* silencing does not provide epithelial cancer cells with a selective growth advantage [17]. A number of groups have reported the relationship between NME1 expression and patient prognosis in NSCLC with different results. Some groups have reported no association between NME1 expression and overall survival [28,29]. In contrast, reduced NME1 expression has been found to be associated with bone metastasis and poor survival in patients with pulmonary adenocarcinoma [30]. Ohta et al. [31] have also reported that NME1 expression is inversely correlated with the microdissemination of tumor cells in stage I NSCLC. In addition, stage I NSCLC patients with NME1-negative expression show a significantly poorer survival than those without [32,33]. The present study also showed the negative effect of reduced NME1 expression on RFS in early stage NSCLC, consistent with findings from previous groups [30,31,32,33].

The negative effect of reduced NME1 expression on RFS in this study was worse in patients with β-catenin overexpression than in those without. How does β-catenin overexpression influence the effect of reduced NME1 expression on RFS? Mechanisms underlying the metastasis suppression of NME1 have been addressed in multiple types of cancer cells. Early efforts have revealed that NME1 may mediate its inhibitory effects on cellular motility and invasion through interactions with signaling cascades [16,34,35,36]. For example, NME1 negatively regulates Rac1 (Rac family small GTPase 1) and Cdc42 (Cell division cycle 42) GTPase by interacting with Rac1-specific nucleotide exchange factors, TIAM Rac1 associated GEF1 (Tiam1) and TGF_BETA_2 domain-containing protein (Dbl-1), respectively [36]. A splicing variant of *NME1* inhibits the metastasis of lung cancer cells by interacting with Inhibitor of nuclear factor Kappa-B Kinase subunit beta (IKKβ) in an isotype-specific fashion and regulating tumor necrosis factor alpha (TNFα)-stimulated Nuclear Factor kappa-light-chain-enhancer of activated B cells (NF-κB) signaling negatively [16]. In addition, NME1 inhibits the liver metastasis of colon cancer cells by regulating the phosphorylation of myosin light chains in nude mice [35]. *NME1* silencing induces the nuclear translocation of β-catenin by disrupting adherence junction complexes mediated by E-cadherin and promotes extracellular matrix invasion by increasing invadopodia formation and pericellular matrix metalloproteinase (MMP) activity [17].

In addition to its effect on signaling pathways, NME1 is also known to regulate gene transcription by binding to single-stranded DNA. A promoter region between –922 to –846 of Kangai 1 (KAI1) is known to suppress metastasis through the inhibition of cell movement. It responds to NME1 in high-metastatic lung cancer cell line L9981 [37]. NME1 suppresses motile and invasive phenotypes of melanoma cells by inducing the transcription of integrin beta-3 (*ITGβ3*) gene through direct physical interaction with the promoter [38]. NME1 also plays a role as a co-regulator of transcription by regulating expression of metastasis-related genes through direct or indirect interactions with transcription-regulatory elements [39,40,41,42]. However, the present study showed no relationship between NME1 expression and nuclear β-catenin expression (Figure 5), suggesting that the adverse effect of NME1 on RFS exacerbated by β-catenin overexpression might not be due to the nuclear translocation of β-catenin by reduced NME1. It is likely that NME1 may interact with β-catenin through other mechanisms such as upregulation of many genes related to cell cycle, apoptosis, and metastasis.

Platinum derivatives such as cisplatin are widely used chemotherapeutic agents for NSCLC. However, cisplatin resistance is a major challenge in the use of these drugs. The molecular mechanism of cisplatin resistance in lung cancer cells is not fully understood. Therefore, there are few efficient strategies to overcome such resistance. Cisplatin-based adjuvant chemotherapy in the present study did not affect RFS in univariate analysis. However, it worsened the RFS in patients with reduced NME1 expression (Appendix A). A functional link between NME1 expression and responsiveness to cisplatin-based adjuvant chemotherapy has been reported by several groups. Cisplatin increases interstrand DNA cross-links and inhibits pulmonary metastatic colonization in *NME1*-transfected breast cancer cells [43]. NME1 has 3′-5′ exonuclease activity potentially involved in DNA proofreading [44]. Thus, reduced expression of NME1 may contribute to chemoresistance by allowing metastatic cells to escape from apoptosis. Knockdown of *NME1* by shRNA transfection in head and neck squamous carcinoma cells attenuates the chemosensitivity of cells to cisplatin by downregulating cyclins E and A and reducing cisplatin-induced S-phase accumulation [45]. These lines of evidence suggest that reduced NME1 expression might be involved in cisplatin resistance through various mechanisms. Therefore, restoring NME1 expression might be a therapeutic intervention strategy to surmount cisplatin resistance.

Previous studies have demonstrated Wnt/β-catenin-mediated resistance to cisplatin in various types of cancers [23,24]. Transient interference of cytoplasmic GSK-3β increases cisplatin resistance by activating Wnt/β-catenin signaling in cisplatin-resistant A549 cells [23]. Recently, Zhang and colleagues [24] have reported that the interference of β-catenin expression by siRNA can decrease mRNA and protein levels of anti-apoptotic gene *Bcl-xl* and increase cisplatin sensitivity in A549 wild-type cells. Despite these associations of β-catenin overexpression and cisplatin resistance in various types of cancer cells, β-catenin overexpression alone was not associated with cisplatin resistance in the present study. However, β-catenin overexpression aggravated RFS when patients with reduced NME1 received cisplatin-based adjuvant chemotherapy (Table 1). Further studies are needed to better understand the combined effect of β-catenin and NME1 on RFS of patients receiving platinum-based adjuvant chemotherapy in early stage NSCLC.

This study was limited by several factors. First, this was a retrospective study that was prone to selection and surveillance biases. Second, it is necessary to investigate combined effects of β-catenin and NME1 on apoptosis, migration, invasion, or metastasis in different cell types of lung cancer to clearly understand the molecular mechanisms underlying the effect of β-catenin and NME1 on poor RFS. Third, the lack of a negative effect of β-catenin in the univariate analysis (Appendix A) in this study might be due to the small sample size and short duration of follow-up. Fourth, *EcoR1* (rs34214448-G/T) polymorphism in *NME1* gene is associated with increased susceptibility to NSCLC [46] and could potentially affect the results of the current analysis, which is based only on expression levels. Fifth, the relationship between the Th1 (T helper cell type 1) and Th 2 (T helper cell type 2) ratio and β-catenin levels were not analyzed in this study. The balance between Th1 and Th2 in the tumor microenvironment is regulated by several factors, and β-catenin may affect the tumor microenvironment. Thus, for the understanding of their relationship and the analysis of β-catenin levels, it may be informative to know the Th1/Th2 ratio of patients. Sixth, the number of patients receiving adjuvant chemotherapy was too small. Accordingly, prospective large-scale studies are needed to validate the effect of β-catenin and cisplatin-based adjuvant chemotherapy on NME1-related RFS in early stage NSCLC.

In conclusion, the present study suggests that the adverse effect of reduced NME1 expression on RFS may be exacerbated by cisplatin-based adjuvant chemotherapy and β-catenin overexpression through other mechanisms rather than through the nuclear translocation of β-catenin in early stage NSCLC. Accordingly, it is recommended that cisplatin-based adjuvant chemotherapy in patients with completely resected stage I–IIIA NSCLC be carefully applied after examining the expression levels of β-catenin and NME1.

## Figures and Tables

**Figure 1 jcm-09-03067-f001:**
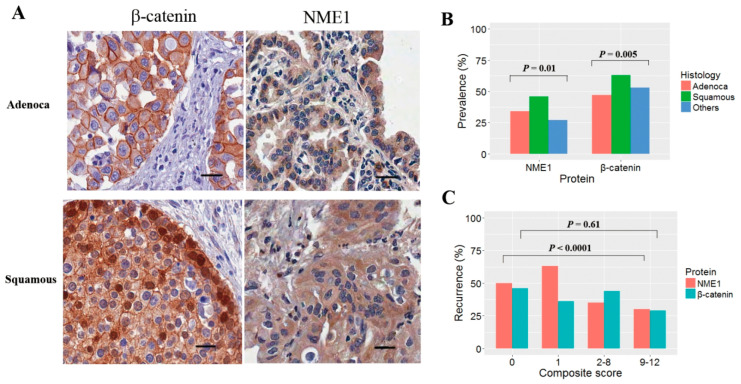
Immunohistochemical staining for non-metastatic cells 1 (NME1) and β-catenin expression in non-small cell lung cancer. (**A**) Expression levels of NME1 and β-catenin were analyzed using immunohistochemical staining (scale bar = 100 μm). Representative images of positive staining are shown in adenocarcinoma (upper) and squamous cell carcinoma (lower) at magnification of ×200. Cytoplasmic staining was considered positive for NME1 and β-catenin expression. (**B**) Prevalence of reduced NME1 expression and β-catenin overexpression was compared according to histologic subtypes. *p-*values were based on Pearson’s chi-square test. (**C**) Association between recurrence and the expression levels of NME1 or β-catenin was analyzed in 425 participants.

**Figure 2 jcm-09-03067-f002:**
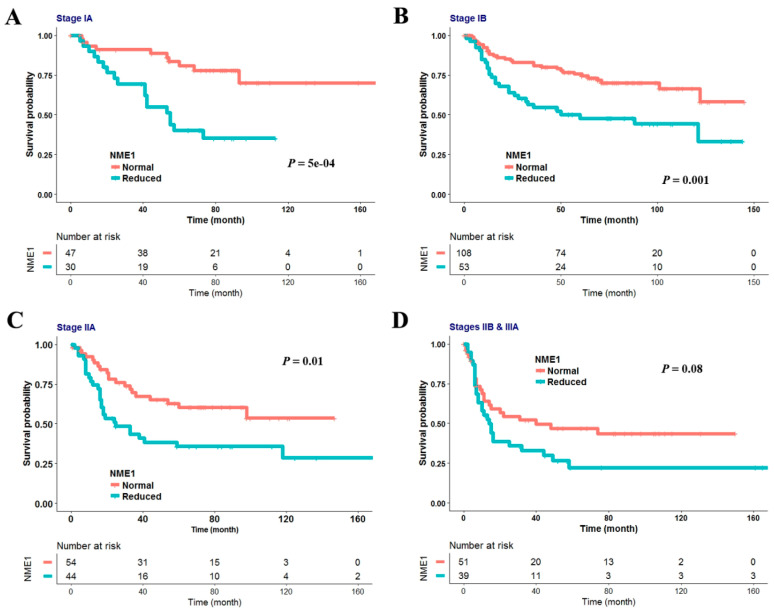
Impact of NME1 on recurrence-free survival (RFS) according to pathologic stages. The effect of reduced NME1 expression on RFS was estimated using the Kaplan–Meier survival curve in 77 patients with stage IA (**A**), 161 with stage IB (**B**), 98 with stage IIA (**C**), and 89 stage IIB–IIIA (**D**). Statistical difference between two survival curves was calculated using the log-rank test.

**Figure 3 jcm-09-03067-f003:**
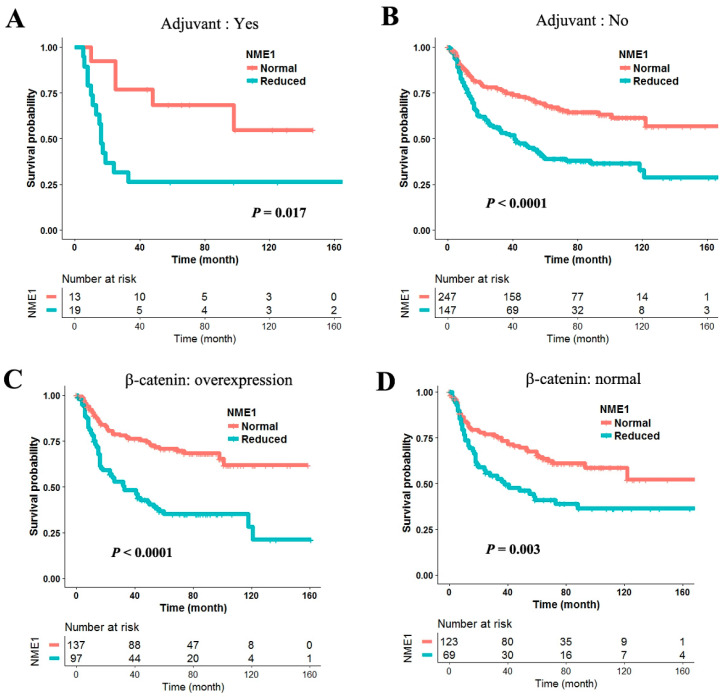
Effect of NME1 on recurrence-free survival, stratified by β-catenin expression and cisplatin-based adjuvant chemotherapy. To understand whether the effect of NME1 on RFS was confounded by β-catenin expression or cisplatin-based adjuvant chemotherapy, data were stratified by β-catenin overexpression (**A**,**B**) or cisplatin-based adjuvant chemotherapy (**C**,**D**) and then the survival curves were compared according to NME1. The survival was compared using the log-rank test in 425 non-small cell lung cancers (NSCLCs).

**Figure 4 jcm-09-03067-f004:**
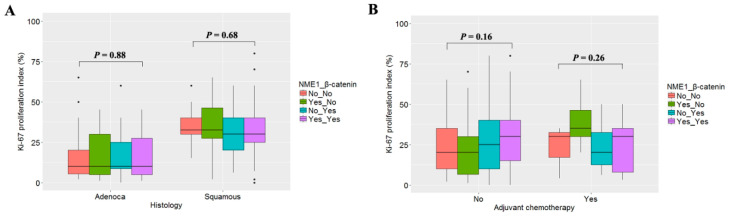
Ki-67 (MKI67) proliferation index according to altered expression of NME1 and β-catenin. Ki-67 proliferation was compared according to expression statuses of NME1 and β-catenin, stratified by histology (**A**) and adjuvant chemotherapy (**B**). “Adenoca” and “Squamous” represent adenocarcinoma and squamous cell carcinoma, respectively. “No” and “Yes” indicate the absence and presence of altered expression, respectively. *p*-values were calculated using one-way ANOVA.

**Figure 5 jcm-09-03067-f005:**
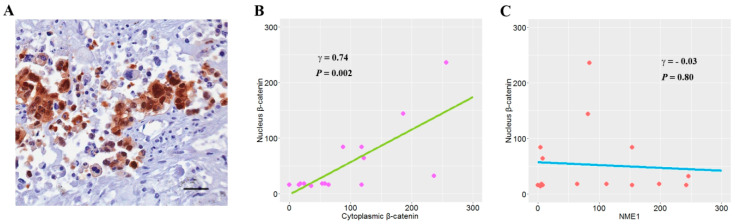
The relationship between NME1 and nuclear β-catenin expression. (**A**) β-catenin expression (scale bar, 100 μm) are shown in nucleus. (**B**,**C**) X- and *Y*-axis scores were obtained by multiplying the intensity score of staining with the proportion of positive stained cells. The linear relationship between two variables was calculated using Pearson’s correlation coefficient.

**Table 1 jcm-09-03067-t001:** Cox proportional hazards analysis ^a^ of RFS according to NME1 in early stage NSCLC stratified by cisplatin-based adjuvant chemotherapy and β-catenin overexpression

Adjuvant Chemotherapy	β-Catenin	Reduced NME1	HR	95% CI	*p*-Value
	Overexpression	Expression			
No	No	No	1		
	(*N* = 176)	Yes	1.89	1.21–2.95	0.005
	Yes	No	1		
	(*N* = 217)	Yes	2.54	1.66–3.89	<0.0001
Yes	No	No	1		
	(*N* = 16)	Yes	5.59	0.62–50.91	0.13
	Yes	No	1		
	(*N* = 16)	Yes	15.52	2.94–82.38	0.001

^a^ Adjusted for age, sex, histology, and pathologic stage. Abbreviations: HR, hazard ratio; CI, confidence interval.

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
