# Peer review of "Negative Effect of Reduced NME1 Expression on Recurrence-Free Survival in Early Stage Non-Small Cell Lung Cancer"

_jcm, 2020, doi:10.3390/jcm9103067_

Round 1
Reviewer 1 Report
In the manuscript Kim et al. the authors analyzed the expression levels of NME1 and β-catenin by immunohistochemistry in tissue from 425 early-stage NSCLC patients. They determined cutoff values for abnormal expression using as control 23 normal lung tissues samples. Successively they analyzed whether cisplatin-based adjuvant chemotherapy and β-catenin over expression can affect NME1 effect on recurrence free survival.
my comments are:
1) how were patients with co-morbidities treated?
2) Please add in the materials and methods section a description for the Ki67 staining.
3) β-catenin levels may affect tumor microenvironment. as suggestion could be informative to know the ratio Th1/Th2 of the patients.
4) it may be with mentioning that the NME1 polymorphism rs34214448 seems to be associated with increased incidence of NSCLC and could potentially also affect the results of the current analysis which is based only on expression levels (doi:10.1016/j.prp.2018.02.020)
Reviewer 2 Report
The authors analysed the effect of NME1 on recurrence-free survival (RFS) in NSCLC patient. Morever they focused on the role of cisplatin based chemotherapy and B-Catenin and WNT pathway activation.
The study demonstrates trough good immunohistochemistry and strong statystical analysis a negative effect of the reduced NME1 expression on RFS in early-stage NSCLC, as already demonstrate by other studies.
Moreover the authors demonstrates a correlation between the negative effect of reduced NME1 expression on RFSand the increased levels of B-Catenint with consequent activation of WNT pathway.The negative effect of reduced NME1 expression on RFS was worse in patients treated with cisplatin based chemotherapy even if this kind of therapy alone does not affect the RFS in this study.
The article is well structured and acceptable with some little clarification and suggestions.
- Please, if possible indicate that the study is a retrospective study also in the matherial and method section.
- Lane 372, please clarify to this reviewer what means the "lack of positive effects of B-Catenin on RFS". B-Catenin overexpression does not exacerbate the negative effects of NME1 reduced levels ?
- Please include the number of patients treated with adjuvant chemotherapy in the matherial and method section.
Best Regards.
